Investigation of microbial community interactions between Lake Washington methanotrophs using ­­­­­­­genome-scale metabolic modeling

Islam Mohammad Mazharul 1
Le Tony 2
Daggumati Shardhat R. 1
Saha Rajib rsaha2@unl.edu 1
1 Department of Chemical and Biomolecular Engineering, University of Nebraska—Lincoln , Lincoln , NE , United States of America
2 Department of Biochemistry, University of Nebraska—Lincoln , Lincoln , NE , United States of America
Chistoserdova Ludmila
Electronic publication date: 2020 Jun 30
Publication date: 2020
Volume: 8
Electronic Location ID: e9464
Received 2020 Feb 21; Accepted 2020 Jun 10
Copyright: ©2020 Islam et al.
Copyright year: 2020
Copyright holder: Islam et al.
License: This is an open access article distributed under the terms of the Creative Commons Attribution License, which permits unrestricted use, distribution, reproduction and adaptation in any medium and for any purpose provided that it is properly attributed. For attribution, the original author(s), title, publication source (PeerJ) and either DOI or URL of the article must be cited.
License URL: https://creativecommons.org/licenses/by/4.0/

Keywords: Methylomonas sp. LW13, Microbial community, Genome-scale metabolic modeling, Methane-cycling community, Lake Washington, Methylobacter tundripaludum 21/22

Funding: University of Nebraska-Lincoln Faculty Startup Grant 21-1106-4308 University of Nebraska-Lincoln UCARE Program This work was supported by University of Nebraska-Lincoln Faculty Startup Grant (21-1106-4308) to Rajib Saha and University of Nebraska-Lincoln UCARE Program. The funders had no role in study design, data collection and analysis, decision to publish, or preparation of the manuscript.

==============================
Background

The role of methane in global warming has become paramount to the environment and the human society, especially in the past few decades. Methane cycling microbial communities play an important role in the global methane cycle, which is why the characterization of these communities is critical to understand and manipulate their behavior. Methanotrophs are a major player in these communities and are able to oxidize methane as their primary carbon source.

Results

Lake Washington is a freshwater lake characterized by a methane-oxygen countergradient that contains a methane cycling microbial community. Methanotrophs are a major part of this community involved in assimilating methane from lake water. Two significant methanotrophic species in this community are Methylobacter and Methylomonas. In this work, these methanotrophs are computationally studied via developing highly curated genome-scale metabolic models. Each model was then integrated to form a community model with a multi-level optimization framework. The competitive and mutualistic metabolic interactions among Methylobacter and Methylomonas were also characterized. The community model was next tested under carbon, oxygen, and nitrogen limited conditions in addition to a nutrient-rich condition to observe the systematic shifts in the internal metabolic pathways and extracellular metabolite exchanges. Each condition showed variations in the methane oxidation pathway, pyruvate metabolism, and the TCA cycle as well as the excretion of formaldehyde and carbon di-oxide in the community. Finally, the community model was simulated under fixed ratios of these two members to reflect the opposing behavior in the two-member synthetic community and in sediment-incubated communities. The community simulations predicted a noticeable switch in intracellular carbon metabolism and formaldehyde transfer between community members in sediment-incubated vs. synthetic condition.

Conclusion

In this work, we attempted to predict the response of a simplified methane cycling microbial community from Lake Washington to varying environments and also provide an insight into the difference of dynamics in sediment-incubated microcosm community and synthetic co-cultures. Overall, this study lays the ground for in silico systems-level studies of freshwater lake ecosystems, which can drive future efforts of understanding, engineering, and modifying these communities for dealing with global warming issues.

Introduction

The accelerated rise in worldwide average temperature in recent years is posing a serious threat to the environment, terrestrial ecosystems, human health, economy, and the ultimate survival of the planet earth. About 20 percent of global warming is caused by methane and it is expected to be 86 times more potent than carbon di-oxide in warming the earth over the next two decades (Houghton, Jenkins & Ephraums, 1990; IPCC, 2013). The impacts of the rapid increase in atmospheric methane (Nisbet et al., 2019) are compounded as higher temperatures are associated with an increase in methane production from wetlands and lakes (Yvon-Durocher et al., 2014). Aerobic methanotrophs are mostly gram-negative Proteobacteria that are an integral part of the global carbon cycle (Hanson & Hanson, 1996). They exist in diverse environments such as wetlands, lakes, and the tundra and use the enzyme methane monooxygenase (MMO) to oxidize methane as their sole source of carbon (Hanson & Hanson, 1996). Methane-oxidizing Verrucomicrobia found in geothermal acidic environments are also involved in methane oxidation while solely using carbon dioxide in the Calvin cycle (Carere et al., 2019; Mohammadi et al., 2019; Van Teeseling et al., 2014). The anaerobic methanotrophic archaea are also an important sink of methane that couple methane oxidation with sulfate reduction mediated by sulfate-reducing bacteria (Cui et al., 2015). Thus, methanotrophs act as the primary biological sink for methane (Hanson & Hanson, 1996), consuming up to 90 percent of the methane produced in soil/sediments in addition to the atmospheric methane (Whalen & Reeburgh, 1990; Krause et al., 2017). Methanotrophs have also shown the ability to produce various useful products such as single-cell proteins, biodiesels, biopolymers, and osmo-protectants (Strong, Xie & Clarke, 2015).

Lakes act as major sources and sinks for methane and account for 6 to 16 percent of biologically produced methane (IPCC, 2013; Yvon-Durocher et al., 2014). Lake Washington is a freshwater lake characterized by a methane-cycling community where methanotrophs are one of the important functional microbial groups involved in methane oxidation (Yu et al., 2016). It contains a steep vertical counter-gradient of methane and oxygen, and is separated into oxic and anoxic layers where methane production and consumption occur, respectively (Yu et al., 2016). Hence, it can be a model system to better understand methane-cycling communities in lakes and their role in the global methane cycle. Understanding the metabolic interactions in these communities will aid in developing methods to reduce the amount of methane emitted from lakes. A diverse array of microbes exist in the Lake Washington community, where Proteobacteria comprise 33% of the community and includes a major subtype of methanotrophs, Methylococcaceae at a 10% abundance level (Beck et al., 2013). The genus Methylobacter is the most dominant player in Methylococcaceae group at 47.7%, while other major players include Crenothrix, Methylomonas, and Methylomicrobium at 30.0%, 10.8%, and 7.4%, respectively (Beck et al., 2013). Other members include cyanobacteria, bacteriodetes, acidobacteria, and chloroflexi (Beck et al., 2013).

Understanding the physiological dynamics and interactions in natural methane-cycling communities, such as the Lake Washington, are crucial to addressing problems concerning methane’s role in global warming and leveraging methanotrophs’ possible functions in bioremediation and bioproduction. Omics-based techniques and high-throughput sequencing can elucidate important features of the community such as taxonomic information, community composition, and presence of functionally important genes (Temperton & Giovannoni, 2012). However, it is difficult to assign functionality to members of the community and decipher the roles of individual players due to the complexity of the community and the data involved (Zengler, 2009; Zengler & Palsson, 2012). On the other hand, synthetic communities were proven to be efficient models to provide insight into metabolic capabilities and interactions (De Roy et al., 2014). Simple representative community structures can be made to lower complexity, achieve more consistent results, and efficiently elucidate inter-species interactions (De Roy et al., 2014). Certain Lake Washington community members are easier to cultivate in a laboratory setting than other members. For instance, Methylomonas and Methylosinus species previously shown ease of cultivation in the laboratory (Auman et al., 2000). However, Methylobacter species were difficult to isolate and demonstrated poor growth compared to Methylomonas and Methylosinus (Yu et al., 2016). A synthetic community comprising 50 different Lake Washington microbes belonging to 10 methanotrophic, 36 methylotrophic, and 4 non-methanotrophic heterobacteria showed that Methylobacter was outperformed by Methylomonas in the community (Yu et al., 2016). Similar results about the dominance of Methylomonas in pure cultures and in standard conditions were observed in their later experiments (Yu, Beck & Chistoserdova, 2017). These observations were inconsistent with previous stable isotope probing studies that found Methylobacter is the dominant Methylococcaceae species among microbes from Lake Washington when grown on methane (Beck et al., 2013; Kalyuzhnaya et al., 2008). These inconsistencies indicate that the complexities of biological systems often make it challenging to understand the functions and interactions within and among organisms in synthetic communities via in vitro and in vivo studies.

In silico evaluation and analysis utilizing mathematical relation-based modeling allow for a high-resolution understanding of the biological processes in a microbial community. The availability of genome-scale metabolic network models combined with biological constraints provide multiple methods to analyze, perform in silico experiment, develop hypotheses, and redesign biological systems at a genome-level (Zomorrodi & Maranas, 2012; Zomorrodi, Islam & Maranas, 2014; Maranas & Zomorrodi, 2016; Islam & Saha, 2018; Alsiyabi, Immethun & Saha, 2019; Islam et al., 2020; Schroeder, Harris & Saha, 2020; Schroeder & Saha, 2019). To develop effective multi-species community models, significant and comprehensive knowledge of inter-species interactions and experimental data must be utilized. Compartmentalized community metabolic models were used to model simple microbial consortia involved in bioremediation, synthetic auxotrophic co-growth, human gut microbiome, and soil bacterial ecosystems (Stolyar et al., 2007; Bizukojc et al., 2010; Lewis et al., 2010; Zhuang et al., 2011; Shoaie & Nielsen, 2014; Henry et al., 2016). On the other hand, community modeling frameworks incorporating the trade-offs between species- and community-level fitness successfully modeled steady-state and dynamic behavior in naturally occurring and synthetic soil microbial communities, synthetic co-cultures for bioproduction, human gut microbiome, and very recently to understand the microbial interactions in bovine rumen and viral influences (Zomorrodi & Maranas, 2012; Zomorrodi, Islam & Maranas, 2014; Chan, Simons & Maranas, 2017; Islam et al., 2019). Other novel methods were also proposed for modeling of such communities involving elementary mode analysis, evolutionary game theory, nonlinear dynamics, and stochastic processes (Vallino, 2003; Lehmann & Keller, 2006; Shou, Ram & Vilar, 2007; Borenstein & Feldman, 2009; Freilich et al., 2009; Frey, 2010; Nadell, Foster & Xavier, 2010; Magnúsdóttir, Heinken & Kutt, 2017). While there have been multiple studies in recent years involved in the model development of various methanotrophs (Akberdin et al., 2018; Bordel et al., 2019a; Bordel, Rojas & Munoz, 2019b; De la Torre et al., 2015; Lieven et al., 2018; Naizabekov & Lee, 2020), an integrated community level analysis of freshwater methane utilizing ecosystems have not been performed yet.

In this work, we developed a simplified community metabolic model with two representative and functionally important strains of Lake Washington, namely, Methylobacter tundripaludum 21/22 (hereafter, Methylobacter) and Methylomonas sp. LW13 (hereafter, Methylomonas) as representative organisms of the methane-oxidizing microbes in the Lake Washington ecosystem. These species were chosen because of their availability in Lake Washington sediments, the ability to mitigate common pollutants, and produce desirable biological products, and the availability of genome-annotation. Draft models of these species were reconstructed followed by careful curation to ensure proper representation of the species. Metabolic pathways and individual reactions that are fundamental to the growth of these organisms such as the ribulose monophosphate pathway, pentose phosphate pathway, methane metabolism, amino acid synthesis and utilization, serine cycle, and Coenzyme B12 biosynthesis were manually scrutinized and then integrated into the models. The metabolites exchanged by the two species models were established by referring to literature and known transporter information (Boden et al., 2011; Caspi et al., 2016; Elbourne et al., 2017; Henry et al., 2010; Kanehisa, 2008; Nguyen et al., 2018; Orata, Kits & Stein, 2018; Svenning et al., 2011; Szklarczyk et al., 2017; UniProt Consortium, 2018; Wartiainen et al., 2006). The curated models of Methylobacter (704 genes, 1,329 metabolites, and 1,404 reactions) and Methylomonas (658 genes, 1,378 metabolites, and 1,391 reactions) were then utilized to develop a community model using a multi-level optimization framework, which was used to estimate biologically feasible metabolite secretion profiles and community compositions (Zomorrodi & Maranas, 2012; Zomorrodi, Islam & Maranas, 2014; Islam et al., 2019). The community was placed under carbon, oxygen, and nitrogen-limiting as well as nutrient-rich environments to study the changes in intracellular carbon and nitrogen metabolism and metabolite excretion profiles. The community composition of carbon-limited environments predicted a shift in the carbon metabolism of both species. The community also demonstrates conservative metabolism under oxygen and carbon-limited environments and produce less carbon-di-oxide. Under these conditions, the mutualistic behavior involving formaldehyde transfer between Methylobacter and Methylomonas is rarely observed. Our results also indicate metabolic reprogramming in TCA cycle and pyruvate metabolism, which can help generate new hypotheses for in vivo experiments. We also simulated the observed binary compositions in a sediment-incubated community and a synthetic co-culture to predict the changes in intra- and extracellular metabolic fluxes. Overall, our results enhance the mechanistic understanding of the Lake Washington methane-cycling community, which can drive further engineering efforts for efficient rerouting of carbon and nitrogen as well as mitigation of methane emission from freshwater ecosystems globally.

Materials & Methods

Metabolic model reconstruction

The draft genome-scale metabolic models of Methylobacter and Methylomonas were developed and downloaded using the ModelSEED database (Henry et al., 2010). The models included reactions for glycolysis/gluconeogenesis, citrate cycle, pentose phosphate pathway, steroid biosynthesis, nucleotide metabolism, and various amino acid biosynthesis. Flux Balance Analysis (FBA), a mathematical approach for analyzing the flow of metabolites through a metabolic network, was utilized for model testing and analyzing flux distributions throughout the work (Orth, Thiele & Palsson, 2010). FBA implements the following optimization framework. 

Maximizevjvbiomass

subject to

(1) ∑j∈JSij.vj=0∀i∈I

(2) LBj≤vj≤UBj∀j∈I

In the framework, I and J represent the sets of metabolites and reactions in the metabolic model, respectively. Sij represents the stoichiometric coefficient of metabolite i in reaction j. The flux value of each reaction j, vj, must be within the parameters of the minimum, LBj, and maximum, UBj, biologically allowable fluxes. vbiomass is the flux of the biomass reaction which simulates the yield of cellular growth in the model (Orth, Thiele & Palsson, 2010).

The biomass composition from recently published methanotroph model Methylomicrobium buryatense strain 5G(B1) (De la Torre et al., 2015) was adopted in this study with slight modification in the lipid macromolecular stoichiometry to account for different lipid macromolecules in the models (see Data S1). Non-growth associated ATP maintenance flux was set to 21.6 mmol/gDCW.hr according to calculated values in the closely related organism Methylomicrobium alcaliphilum by Akberdin et al. (2018). All of the three modes of electron transfer during Methane oxidation (redox arm, direct coupling and uphill electron transfer) have been included in the models, since they are all possible and there is no definitive conclusion of which organism prefers what mode of methane oxidation.

Metabolic model curation

Draft models for both Methylobacter and Methylomonas underwent an extensive manual curation process, including chemical and charge-balancing, elimination of thermodynamically infeasible cycles, and ensuring network connectivity. Reactions in the draft model reaction set imbalanced in protons were checked with their appropriate protonation states and corrected by adding and deleting proton(s) on either side of the reaction equation. The remaining imbalanced reactions were stoichiometrically inaccurate and required the atoms on both sides of the reaction equation to be balanced. The metabolites consumed and produced by models were established by referring to literature and known transporter information (Boden et al., 2011; Caspi et al., 2016; Elbourne et al., 2017; Henry et al., 2010; Kanehisa, 2008; Nguyen et al., 2018; Orata, Kits & Stein, 2018; Svenning et al., 2011; Szklarczyk et al., 2017; UniProt Consortium, 2018; Wartiainen et al., 2006).

Genome annotations of methanotrophs indicate that formaldehyde assimilation can happen through the oxidative Pentose Phosphate Pathway as well as the formaldehyde/formate dehydrogenases and the tetrahydrofolate-associated pathways (Cai et al., 2016; Khmelenina et al., 2018). However, there have been extensive evidences that the Ribulose Monophosphate (RuMP) cycle is the major route for formaldehyde assimilation (Fu, Li & Lidstrom, 2017; He et al., 2020; Peyraud et al., 2011). Also, multiple studies indicate that formaldehyde assimilation through the Serine cycle is insignificant (De la Torre et al., 2015; Fu, Li & Lidstrom, 2017; He et al., 2020). Based on transcriptomic data and metabolic flux measurements (De la Torre et al., 2015; Fu, Li & Lidstrom, 2017), we restricted the distribution of formaldehyde assimilation between the RuMP cycle and the tetrahydrofolate-associated pathways. In addition, the oxygen stoichiometry in pathways related to direct coupling methane oxidation and the coupling with cytochrome c oxidase have been set (1 mol Oxygen/mol pyrroloquinoline quinone and 0.5 mol Oxygen/mol Cytochrome c) according to literature (De la Torre et al., 2015; Fu, Li & Lidstrom, 2017; Lieven et al., 2018; Sugioka et al., 1988).

While applying mass balance constraints to genome-scale metabolic models can display the net accumulation and consumption of metabolites within each microbial model, it fails to account for the regulation of reaction fluxes. The limitation of this constraint is better elucidated when focusing on reaction cycles that do not consume and produce metabolites. Because of the absence of metabolite consumption and production, the overall thermodynamic driving force of the cycles become zero and the cycle is incapable of supporting any net flux, and thus deemed thermodynamically and biologically infeasible (Schellenberger, Lewis & Palsson, 2011). These thermodynamically infeasible cycles in our models were identified by inhibiting all nutrient uptakes to the cell and utilizing the optimization framework, Flux Variability Analysis (FVA), which maximizes and minimizes each reaction flux within the model based on the mass balance constraints (Mahadevan & Schilling, 2003). Reactions whose fluxes reached the defined lower and upper bounds were determined to be unbounded reactions, group together based on stoichiometry, and systematically corrected. These cycles were eliminated by removing duplicate reactions, turning off lumped reactions, fixing reaction directionality, or selectively turning reactions on or off based on cofactor specificities found from literature (Ishida et al., 1969; Dekker, Lane & Shapley, 1971; Chen, Lee & Chang, 1991; Schomburg & Stephan, 1995; Achterholt, Priefert & Steinbuchel, 1998; Hadfield et al., 1999; Hutter & Singh, 1999; Kai, Matsumura & Izui, 2003; Feist et al., 2007; Dean, 2012; Flamholz et al., 2012; Lin et al., 2014; Christensen et al., 2017). The FVA optimization algorithm is shown below.

Maximize∕minimizevjvj

subject to

∑j∈JSij.vj=0∀i∈I1

LBj≤vj≤UBj∀j∈I2

(3) vbiomass=vapp−obj,threshold

Both metabolic models were checked for the ability to produce biomass and metabolites they were known to produce (Svenning et al., 2011; Kalyuzhnaya et al., 2015). The metabolic functionalities of the models were ensured by identifying and manually adding reactions from biochemical databases, such as KEGG (Kanehisa & Goto, 2000) and Uniprot (UniProt Consortium, 2018), to each model. Fully developed models of related organisms such as Methylococcus capsulatus and Methanomonas methanica for both Methylobacter and Methylomonas were utilized in help pinpoint absent enzymatic activity within our models. The addition of these reactions was confirmed using the bioinformatic algorithm, BLAST, which compares the genes of our organisms and related organisms and determines whether they are found to be orthologous. It was then tested whether these reactions would increase the number of thermodynamically infeasible cycles and promote for further curation of the models. 

Community model formation

Following the curation of each individual microbial model, both were implemented to form a community model using the bi-level multi-objective optimization framework OptCom (Zomorrodi & Maranas, 2012). OptCom simultaneously optimizes each individual community member’s flux balance analysis problem as an inner-level optimization problem and the community model objective as an outer-level optimization problem. The maximization of the community biomass was set as the community objective function. The mathematical framework of the OptCom procedure is the following. 

Maximizeorminimizez=Communitylevelobjective

subject to,

maximizevjkvbiomassksubject to,∑j∈JkSijk.vjk=0∀i∈Ik1LBjk≤vjk≤UBjk∀j∈Jk2vuptake,ik=ruptake,ik∀i∈Iuptakek4vexport,ik=rexport,ik∀i∈Iexportk5∀k∈K

∑kruptake,ik+eic= ∑krexport,ik+uic∀i∈I shared6

ruptake,ik,r export,ik,eic,uic≥0∀i∈I shared,∀k∈K

The inner-level optimization problem(s) represents the steady-state FBA problem for each community member k and limits the exchange (uptake and export) fluxes of shared metabolites between the individual species using the outer-level optimization problem parameters ruptake,ik and rexport,ik, respectively. The outer-level optimization problem constraint characterizes the mass balance for every shared metabolite in the extracellular environment within the shared metabolite pool. Metabolic interactions, with constraints, were confirmed with prior experimental research involving both community members or one member with a related organism found in the Lake Washington community (Yu et al., 2016).

Formaldehyde inhibition constraint

Formaldehyde production can be inhibitory towards the growth of Methylomonas and Methylobacter (Bussmann, Rahalkar & Schink, 2006). Methylobacter is able to uptake formaldehyde excreted from Methylomonas to alleviate the inhibitory effects on Methylomonas growth. A constraint based upon the minimum and maximum inhibitory concentrations of formaldehyde was implemented to simulate formaldehyde’s inhibitory effects on Methylobacter (Hou, Laskin & Patel, 1979; Costa, Dijkema & Stams, 2001). While formaldehyde production is not forced in the model simulations, this constraint accounts for the resulting growth inhibition if there is any amount of formaldehyde excreted. The minimum formaldehyde concentration required for growth inhibition was found to be 1 mM (Hou, Laskin & Patel, 1979) and the formaldehyde concentration required for total growth inhibition (maximum formaldehyde concentration) was found to be 7 mM (Costa, Dijkema & Stams, 2001). vbiomassMethylobacter≤v maxbiomassMethylobacter×1−R×v formaldehydeuptakeMethylobacter

The parameter R was derived by finding the rate of change of biomass growth by the change in formaldehyde concentration.

Bridging metabolic network gaps

Automated draft reconstructions are limited as many reaction networks possess gaps due to missing reactions and blocked reactions. These are defined as reactions that lack production/consumption of its reactants/products. Major metabolic pathways were added based upon the literature of each organism (Kalyuzhnaya et al., 2015). Gaps were filled by referencing genetically related organisms to find missing metabolic capabilities. The presence of these possible reactions in the models were validated by cross referencing the relevant amino acid sequences between the reference organism and our models via pBlast. The reactions were then checked for the formation of thermodynamically infeasible cycles before being accepted.

Computational resources

The General Algebraic Modeling System (GAMS) version 24.8.5 with IBM CPLEX solver was utilized to run the optimization algorithms. The optimization frameworks were scripted in GAMS and then run on a Linux-based high-performance cluster computing system at the University of Nebraska-Lincoln. The downloaded models from ModelSEED with curations were parsed from System-Biology Markup Language (SBML) level 2 documentation using general-purpose programming language Python to generate the input files necessary for GAMS and subsequent manual curations.

Results

Genome-scale metabolic models of Methylobacter and Methylomonas

The draft genome-scale models of Methylobacter and Methylomonas are reconstructed using the ModelSEED database (Henry et al., 2016). The manual curation process ensures that there is no chemical and charge imbalance present in either of the models and there is no reaction with unrealistically high fluxes (infeasible reactions) without any nutrient uptake. The manual curation also reconnects a significant number of blocked metabolites to the network in both models (i.e., 107 metabolites for Methylobacter and 109 metabolites for Methylomonas). This enhancement of network connectivity is performed using available literature pertaining to major metabolic pathways that are known to be present in both the microbes (Kanehisa & Goto, 2000; Kalyuzhnaya et al., 2015). The draft models were lacking some reactions in the important metabolic pathways i.e., the methane oxidation, pentose phosphate pathway, nitrogen fixation, cofactor, and amino acid production, which are curated at this stage. The model statistics are shown in Table 1. Data S2 and S3 contain the model files for Methylobacter and Methylomonas, respectively.

Table 1 Model statistics for Methylobacter and Methylomonas.

Models	Methylobacter	Methylomonas	
Genes	704	658	
Reactions	1,404	1,391	
Metabolites	1,329	1,378	
Blocked reactions	660	672	

Community dynamics under variable environmental conditions

The individual models are integrated into a community model using available multi-objective computational optimization framework (Zomorrodi & Maranas, 2012). The metabolic interactions between Methylobacter and Methylomonas are described using inter-organisms flow constraints. The community, as a whole, consumes methane, oxygen, and nitrogen, which is then shared between Methylobacter and Methylomonas. In addition, Methylobacter consumes formaldehyde produced by Methylomonas under certain conditions, which alleviates the formaldehyde toxicity on Methylomonas growth (Bussmann, Rahalkar & Schink, 2006). At the same time, both Methylobacter and Methylomonas export carbon di-oxide to the environment. The community is simulated under four conditions in which methane, oxygen, and nitrogen are supplied to the community at various levels. These conditions are denoted as A, B, C, and D in Table 2 and correspond to the panels in Fig. 1. It should be noted that the amount of each nutrient consumed by the members are not necessarily equal to the amount of nutrient supplied as only one of the nutrients acts as a limiting reagent in each limiting condition. The community model is illustrated in Fig. 1.

Table 2 High and limiting nutrient conditions for community simulation.

Condition	Methane uptake (mmol/gDCW.hr)	Oxygen uptake (mmol/gDCW.hr)	Nitrogen uptake (mmol/gDCW.hr)	
A (high C, high O, high N)	100	100	100	
B (high C, limiting O, high N)	100	50	100	
C (high C, high O, limiting N)	100	100	40	
D (limiting C, high O, high N)	20	100	100	

Figure 1 Community dynamics showing the fluxes of key shared metabolites and biomass in the community model.

(A) High nutrient condition; (B) oxygen limited condition; (C) nitrogen limited condition; and (D) carbon limited condition.

The community biomass flux is the highest when all the nutrients are highly abundant (Fig. 1A). Methylomonas dominates the community in nutrient-rich condition. It is observed that a limited uptake of oxygen also imposes some restriction on the carbon and nitrogen utilization by the community, which results in reduced biomass fluxes for both Methylomonas and Methylobacter (Fig. 1B). When oxygen uptake is limited, methane utilization by Methylomonas is still higher compared to Methylobacter despite an overall decrease in methane consumption by the community. An overall conservative nature of metabolism is observed in the community, as manifested by the least amount of carbon-di-oxide production and no formaldehyde production. In nitrogen-limited growth (Fig. 1C), Methane uptake by Methylomonas decreases further and a high rate of respiration is observed, with the highest production of carbon-di-oxide. In Methane-limited condition (Fig. 1D), while the metabolism in Methylobacter remains mostly unaffected, Methylomonas growth is severely hindered by the scarcity of Methane. In all the nutrient-limited conditions, Methylobacter is observed to dominate the community. The observations from Optcom simulations are consistent with the shifts in possible flux ranges under different conditions, as estimated by Flux Variability analysis. These observations indicate that the inherent degeneracy of metabolic fluxes in Flux Balance Analysis or Optcom does not affect the conclusions obtained in this work. The detailed results are presented in Data S4.

The total community biomass and community composition under methane and oxygen gradients are simulated to model the methane-oxygen counter gradient that exists in Lake Washington (Yu et al., 2016). In general, the total community biomass is observed to increase with both oxygen and methane uptake, as is expected from the increased abundance of nutrients. The community is completely dominated by Methylobacter under low methane/low oxygen conditions. An increase of methane at low oxygen conditions does not change the dominance of Methylobacter. Furthermore, the increase of oxygen under low methane conditions has minimal impact in changing the community composition, as can be seen from very slowly increasing ratio of Methylomonas to Methylobacter across the entire range of oxygen uptake at low methane uptake condition (Fig. 2). The biggest charge in community composition is observed under high carbon and high oxygen conditions in which the community biomass is composed of 32% Methylobacter and 68% Methylomonas, as compared to 99% Methylobacter and 1% Methylomonas in the low methane/low oxygen condition (Fig. 2). The switch in community composition is consistent with the observations in nutrient rich condition. The complete numerical results are included in Data S4.

Figure 2 The community composition and total biomass under varying methane and oxygen conditions.

The size of each pie chart represents the total community biomass.

In nutrient-rich condition (shown in Fig. 3), Methylobacter is the dominant community member and consumes the major portion of all of the shared the metabolites i. e., methane, oxygen, and nitrogen. Methylomonas consumes the major share of methane (80%) and most of it is accumulated in biomass with a small amount of formaldehyde (0.1 mmol/gDC.hr) being produced. Methylomonas also has a more active serine cycle converting formaldehyde into the metabolites in the central carbon metabolism. While the TCA cycle in both Methylobacter and Methylomonas is active in nutrient-rich condition, the activity of alpha-ketoglutarate dehydrogenase was very low.

Figure 3 Central carbon metabolism fluxes in Methylobacter (A) and Methylomonas (B) under high nutrient condition.

Under the oxygen limiting condition (Fig. 4), the flux through the methane oxidation pathway decreases by 20% in Methylobacter. Methylomonas consumes the majority (60%) of the total methane uptaken by the community under the oxygen limiting condition (Fig. 1B). Methylomonas diverts central carbon compounds to produce pyruvate via the assimilation of carbon di-oxide and acetaldehyde under the oxygen limited condition (Fig. 4B). A small fraction of carbon di-oxide downstream of this reaction is secreted into the environment, which is the lowest among all the conditions. In this condition, Methylomonas has a complete TCA cycle activity.

Figure 4 Central carbon metabolism fluxes in Methylobacter (A) and Methylomonas (B) under oxygen limited condition.

The methane oxidizing pathway and TCA cycle reactions in Methylobacter do not show any significant change in nitrogen limited condition (Fig. 5A). Methylobacter excretes more carbon di-oxide than nutrient-rich condition while consuming more oxygen than Methylomonas. On the other hand, the activity of the methane oxidation pathway of Methylomonas decreases (the lowest in any non-carbon limiting conditions). Furthermore, succinyl-CoA in Methylomonas is produced from alpha-ketoglutarate in the TCA cycle, similar to oxygen limited conditions (Fig. 5B). Methylomonas, similar to Methylobacter, also excretes significantly high amount of carbon-di-oxide in nitrogen-limited condition.

Figure 5 Central carbon metabolism fluxes in Methylobacter (A) and Methylomonas (B) under nitrogen limited condition.

In the carbon limited growth condition, Methylomonas takes up very small amount of methane, while Methylobacter takes up most of the methane supplied to the community (Fig. 6). The central metabolism of Methylobacter is not altered under this carbon limited condition (Fig. 6A). However, in Methylomonas, carbon di-oxide is scavenged by assimilating succinyldihydrolipoamide and carbon di-oxide to succinyl-CoA (Fig. 6B). This reaction is inactive in the oxygen limiting condition. Methylomonas also displays minimal activity in its serine cycle under carbon limiting condition.

Figure 6 Central carbon metabolism fluxes in Methylobacter (A) and Methylomonas (B) under carbon limited condition.

Dynamic shifts in metabolism under sediment incubated microcosm and synthetic co-culture composition

Previous studies have shown inconsistencies between the microcosm incubated from Lake Washington sediments and synthetic community cultured in the lab. In the natural community (microcosm incubated from the lake sediments), Methylobacter has been shown to be dominant under both low methane/high oxygen and high methane/low oxygen conditions (Beck et al., 2013). However, Methylomonas outcompeted Methylobacter in both low methane/high oxygen and high methane/low oxygen conditions in synthetic co-cultures (Yu et al., 2016). To elucidate the metabolic flux distributions and the extent of inter-species interactions that gives rise to the observed community composition in the two conditions, the experimentally observed species abundance ratio was imposed on the growth rates of Methylobacter (MB) and Methylomonas (ML) as a constraint in the community optimization framework. For the sediment-incubated community, a MB:ML ratio of 9.3:1.0 and for the synthetic community, a MB:ML ratio of 0.05:1.0 were used.

The vast majority of reactions in Methylomonas has lower flux values in the sediment-incubated community compared to the synthetic community (Fig. 7). This occurs because Methylomonas constitutes a smaller portion of the total community biomass in the sediment-incubated community. There are alternate pathways to produce pyruvate which has increased flux (Fig. 7A). These pathways produce pyruvate by assimilating carbon di-oxide and acetaldehyde and by the assimilation of cysteine and mercaptopyruvate. However, Methylomonas produces less pyruvate overall even with the increased flux in these pathways, since other pyruvate producing pathways decrease in flux. Primarily, the flux through L-serine and ammonia assimilation to produce pyruvate is high in the synthetic condition but decreases in sediment-incubated condition. Under sediment-incubated conditions, Methylobacter uses methane as its primary carbon source and does not consume any formaldehyde produced by Methylomonas. On the other hand, Methylobacter takes up some of the secreted formaldehyde from Methylomonas as a carbon source in addition to methane under the synthetic condition (Fig. 7B).

Figure 7 Flux distribution for select metabolites in Methylobacter and Methylomonas under (A) Lake Washington sediment-incubated microcosm conditions and (B) synthetic co-culture conditions.

Discussion

Aerobic methane oxidation in freshwater lakes around the world is a key metabolic process that significantly affects the carbon cycle by acting as a major sink of methane. Lake Washington provides an wonderful opportunity to study the methane cycling, with up to 20% of the organic carbon being released as methane through decomposition and consuming up to 10% of the dissolved oxygen in the lake water (Kuivila et al., 1988). With the goal to understand the dynamics of the methane cycling Lake Washington community, we integrated high-quality manually curated and refined genome-scale metabolic models of highly abundant, functionally important, and representative community members using multi-level optimization-based frameworks. While this community have been studied by many researchers in the previous years with in vivo tools like synthetic co-cultures, metagenomics, and metatranscriptomics (Hernandez et al., 2015; Oshkin et al., 2015; Yu et al., 2016; Krause et al., 2017), an in silico approach like the one used in this work is needed to have a deeper understanding of the underlying mechanisms that govern the inter-species interactions and in turn, the community structure, function and dynamics in Lake Washington. As the global carbon transactions are changing and the release of greenhouse gases in the atmosphere is consistently getting worse every year, it is imperative for us to put our best efforts in mitigating the harmful effects. To do that, the use of genome-scale metabolic modeling tools to understand how the microbial communities are involved in these metabolic processes function in natural environments is essential.

To make the community model a good representation of the naturally occurring methane cycling community, we selected two highly abundant and functionally important microbial species. Following the manual curation process of both metabolic models, it was found that Methylobacter was more efficient than Methylomonas at producing biomass when simulated under the same standard growth environment and biological constraints in the Lake Washington, except at highly nutrient-rich conditions. Since Methylobacter and Methylomonas are competitors for the sole carbon source, methane, the overall metabolic efficiency is an important factor in the methane utilization ratio of the two species. While there is no direct literature evidence that suggests that one of them is more efficient in utilizing methane for growth compared to the other, it is highly possible that they might be limited by other small molecules that inhibit high methane consumption. For example, the community was tested under carbon, oxygen, and nitrogen limited conditions to observe how its central metabolic pathway and the community composition varied, and we observe the dominance of Methylobacter in the community in all of the limited growth conditions (Fig. 4 through 6) but the dominance of Methylomonas in the nutrient-rich condition. This is observed even during the oxygen limited growth condition, where Methylobacter is unable to consume as much methane as Methylomonas. In this condition, Methylomonas takes up more methane than Methylobacter but Methylobacter can still maintain its dominance.

Each of the nutrient limited conditions shows variable differences within the methane oxidation pathway, the serine cycle, and the TCA cycle. While, based on the simulation, we did not observe a noticeable flux through the serine cycle in either Methylomonas or Methylobacter in all nutrient-limited conditions, it was significantly active in Methylomonas in nutrient rich condition. In oxygen limiting condition, the community was also observed to conserve as much resource as possible. For example, while Methylomonas excretes some amount of carbon dioxide in all conditions, it routes most of it back to central carbon metabolism through the direction reversal of pyruvate decarboxylase. Although there is currently no experimental studies pointing to this phenomenon, studies in other organisms suggest a high oxygen sensitivity of this enzyme (Eram & Ma, 2013). Therefore, a possible explanation of the shifts in pyruvate metabolism is the oxygen sensitivity of this enzyme, which needs to be further studied.

In most of the simulation conditions, Methylobacter and Methylomonas were observed to assimilate most of the consumed carbon though the Entner-Doudoroff (ED) pathway. This was evident from the high flux of fructose-6-phospahte to 6-phophogluconate and thereafter pyruvate. In contrast, the Embden-Meyerhof-Parnas (EMP) variant of the glycolytic pathway was predicted to be the dominant pathway in some other methanotrophs like M. buryatense through model simulations (De la Torre et al., 2015; He, Fu & Lidstrom, 2019; Kalyuzhnaya et al., 2013). It should be noted that studies measuring glycolytic fluxes experimentally in Methylobacter and Methylomonas species are sparse in literature. A recent 13C tracer analysis by Fu, Li & Lidstrom (2017) suggested that the flux partition between EMP and ED variants is unresolved since they both manifest fully labeled downstream molecules. Another interesting observation from this study is the activity levels of alpha-ketoglutarate dehydrogenase enzyme in all conditions. The presence and expression of alpha-ketoglutarate dehydrogenase in methanotrophs has been a matter of debate for quite some time (Zhao & Hanson, 1984; Theisen & Murrell, 2005), which is manifested as the inability of methanotrophs to grow on multi-carbon substrates (Smith, Trotsenko & Murrell, 2010). In this study we observe very negligible (<2% of the TCA cycle flux) alpha-ketoglutarate dehydrogenase activity in all conditions. At this point it is not straightforward to decipher what exactly might be the regulating factor to this enzyme and warrants further experimentation.

Finally, the community model was simulated with fixed abundance ratios of the two members to reflect the composition in synthetic co-culture and sediment-incubated community. The changes in community composition as oxygen and carbon levels change are more consistent with behaviors in sediment samples than synthetic co-cultures (Hernandez et al., 2015; Yu et al., 2016). Methylobacter is able to utilize methane to produce biomass at a more efficient level than Methylomonas. However, when the synthetic co-culture condition is imposed on the species abundance ratio, i.e., favoring Methylomonas biomass, Methylobacter takes up formaldehyde produced by Methylomonas in addition to methane consumption, which allows it to bypass the oxygen-intensive reaction of oxidizing methane to methanol to some degree. This commensal relationship helps Methylobacter to enhance its biomass even when the uptake of the original carbon source (methane) is low while protecting Methylomonas from the toxicity and growth inhibitory effects of formaldehyde (Hou, Laskin & Patel, 1979; Costa, Dijkema & Stams, 2001). In our optimization formulation, the inhibitory formaldehyde constraint placed on Methylobacter makes the consumption of formaldehyde detrimental towards biomass production, but it can simultaneously act as a carbon source and compensates for its inhibitory effects. This is contradictory to what was observed in the co-culture experiments (Yu et al., 2016). Similar to formaldehyde, methanol export was also not observed in the simulations although some previous studies have reported ethanol production(Krause et al., 2017; Zheng et al., 2018). This lack of excretion of intermediate carbon molecules is often observed in steady-state Flux Balance Analysis models without any enzyme kinetic information, where pathways upstream of biomass consumes most of the intermediates. However, it should be noted that, the synthetic co-culture experiments reported in literature involved other community members, whose metabolic interactions with Methylobacter and Methylomonas are not well characterized to date. Some studies have indicated co-operative relationships between Methylobacter and other microbial species from Methylophilaceae family (Beck et al., 2013), which might potentially impact that community dynamics in synthetic co-cultures. Moreover, despite the dominance of Methylomonas in the synthetic community at a species level (Soni et al., 1998; Hoefman et al., 2012), further assessment of community composition at a higher taxonomic level indicated a consistency with naturally observed composition (Yu et al., 2016). We hypothesize that this discrepancy is possible, given that there is high functional redundancy present in Lake Washington community, similar to any naturally occurring microbial ecosystem (Galand et al., 2018; Louca et al., 2018; Islam et al., 2019; Jia & Whalen, 2020).

Conclusion

In this work, we attempted to enhance our mechanistic understanding of the dynamics in the methane cycling Lake Washington community through genome-scale metabolic modeling of the representative and functionally important community members, Methylobacter tundripaludum 21/22 and Methylomonas sp LW13. The understanding of this community behavior will be a foundation for future studies that aim at the long-term goal of creating a complex synthetic community capable of carrying out certain desired functions through the consumption of methane, thus mitigating the harmful effects of methane release in the atmosphere. One should be aware of the fact that the in silico results need to be further tested and confirmed through relevant experiments before any engineering strategies can be successfully employed. The community metabolic model, in that regard, should be expanded to include other major players of the Lake Washington community, i.e., members of Bacteroidetes and Proteobacteria phyla. Including these organisms in our community metabolic model will enable us to explain currently unidentified inter-species metabolite exchanges/interactions that play important role in the cycling of methane as well as other nutrients.

Supplemental Information

Data S1 Biomass composition

Estimated biomass coefficient for each of the biomass precursors.

Click here for additional data file.

Data S2 Methylobacter metabolic model

File in Systems Biology Markup Language (SBML) format, readable by any standard text editor.

Click here for additional data file.

Data S3 Methylomonas metabolic model

File in Systems Biology Markup Language (SBML) format, readable by any standard text editor.

Click here for additional data file.

Data S4 Community simulation results in different conditions

Biomass contributions of Methylobacter (MB) and Methylomonas (ML) across methane and Oxygen gradient, Maximum and minimum flux ranges of Methylobacter and Methylomonas at different nutrient conditions, and Optcom-predicted flux distribution in Methylobacter and Methylomonas at different nutrient conditions.

Click here for additional data file.

Additional Information and Declarations

Competing Interests

Author Contributions

Data Availability

The authors declare there are no competing interests.

Mohammad Mazharul Islam conceived and designed the experiments, performed the experiments, analyzed the data, prepared figures and/or tables, authored or reviewed drafts of the paper, and approved the final draft.

Tony Le performed the experiments, analyzed the data, prepared figures and/or tables, authored or reviewed drafts of the paper, and approved the final draft.

Shardhat R Daggumati performed the experiments, analyzed the data, prepared figures and/or tables, authored or reviewed drafts of the paper, and approved the final draft.

Rajib Saha conceived and designed the experiments, prepared figures and/or tables, authored or reviewed drafts of the paper, and approved the final draft.

The following information was supplied regarding data availability:

The simulation results and model files are available in the Supplementary Files.

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
