# Peer review of "Investigation of microbial community interactions between Lake Washington methanotrophs using ­­­­­­­genome-scale metabolic modeling"

_PeerJ, doi:10.7717/peerj.9464_

## Round 0.1 · original submission · Major Revisions

As you can see the reviewers find your study of interest but express serious concerns with respect to your understanding of metabolism of C1 compounds in Type I methanotrophs. It is the basic knowledge supported by studies published over the past 50 plus years that these organisms utilize the ribulose monophosphate cycle as the main pathway for assimilating formaldehyde. The serine cycle may play an additional role in assimilation or detoxification but cannot be modeled as the main route for building biomass from formaldehyde. Thus, you model is in principle incorrect. In order to continue with this submission, you need to incorporate the ribulose monophosphate cycle into your model as well as address other concerns by the reviewers such as the role of formaldehyde in carbon exchange.

Reviewer 1 ·

Basic reporting

The manuscript is well written, clear and unambiguous. The introduction covered all the content needed to understand the results with the literature well-referenced and current. All figures are in good quality and well described. My concerns and suggestions are described in Comments for the author.

Experimental design

This work is within the scope of the journal and shows a well-designed and rigorous investigation with coherent methods to achieve the objectives. Methods were described with sufficient detail and information to replicate.

Validity of the findings

The microbial community interactions of methanotrophs is of interest to the field, so this study is a welcome addition.

Additional comments

This manuscript by Islam el al describes an effort for developing highly curated genome-scale metabolic models of Washington lake isolating methanotrophs and investigating microbial community interactions between them. The objective was clear and the method applied such as GSM reconstructions, flux analysis, community model formation which revealed some interesting insights into the interactions of methanotrophs in this microbial community. Based on this property the manuscript is of interest to the C1 community. However, in order to be more suitable for publication, the authors are encouraged to address the following concerns:
Major comments:
- The reviewer concerns about how the biomass equation and non-growth associated maintenance of these models were generated. Please include in the Materials and Methods section.
- There were three modes for methane oxidation including redox arm, direct coupling and uphill electrons. Which mode was included in these models of Methylobacter and Methylobacter?
- One significant question the reviewer has is that Methylobacter Tundripaludum 21/22 and Methylomonas sp. LW13 are members of the Gammaproteobacteria (type I methanotroph) so that they should use RuMP cycle as the main pathway for formaldehyde assimilation. Additionally, they only possess an incomplete serine cycle, lacking reactions converting acetyl coenzyme A to glyoxylate. Therefore, the serine cycle cannot serve as the major carbon assimilation pathway in type I methanotrophs. However, as described in the manuscript, the flux analysis showed that all of the consumed methane enters the serine cycle for biomass formation and no carbon goes through RuMP cycle (HPS and PHI fluxes were zero). Please explain about this.
Minor comments:
- Line 131-132: There were several accurate genome-scale metabolic models that have been developed in both type I and type II methanotrophs which improved our knowledge on C1 metabolism. These relevant papers could have been cited in the Introduction section in order to give a better picture of the current state of the art in this field.
- Line 248-249: Please add the references for the Methylococcus capsulatus and Methanomonas methanica models.
- Line 402-404: The author described Methylomonas have branched and a complete oxidative TCA cycle. The reviewer recommends the author to describe TCA cycle in Methylobacter in order to compare the TCA cycle between these methanotrophs under different conditions.

Reviewer 2 ·

Basic reporting

To my knowledge, this is the first study on simulation of a methanotrophic community comprising of two type I methanotrophs. This topic is of great importance since it is experimentally challenging to resolve flux distributions of individual microbes in a bacterial community. This paper has a nice introduction regarding both the methanotrophic communities and simulation of cocultures via genome-scale models. Overall, the manuscript is well-written, and all figures and tables are well described.

Experimental design

Overall, the method and materials part has presented detailed information of how the genome-scale models were built and how the simulation was achieved via multi-layer optimization. I have a few questions:
1. What is the community level objectivity exactly applied for all the growth conditions? Can you clarify that in the method part? Are there multiple community level objectives possible, which might lead to different results?
2. How about the maintenance energy? Can you add some comments on what value(s) you have used? Or did you just let the model find whatever gives the optimal solution? If this is the case, you might need to check if the value is within reported ranges of energy cost by cell maintenance.
3. Methanotrophs are known to produce exopolymeric substances (EPS), especially at a high cell density. Have you considered the EPS formation in the model?

Validity of the findings

With regard the findings, I have some major concerns, because some of them do not make sense for type I methanotrophs:

1. For aerobic methanotrophs, the theoretical O2:CH4 consumption ratio is between 1 and 1.5. The pMMO may oxidize NH4 as well, so this number can be higher than 1.5. However, based on Figure 1 and 7, this ratio is predicted to be higher than 3 in general. This does not make sense. Please see the following research articles which either measured or predicted the O2/CH4 consumption ratios (ranging from 1.0 to 1.6) agreeable with theoretical values.
a. Leak, D. J., Dalton, H., 1986. Growth yields of methanotrophs. Appl. Microbiol. Biotechnol. 23, 470-476.
b. Gilman A, et al. 2015. Bioreactor performance parameters for an industrially-promising methanotroph Methylomicrobium buryatense 5GB1. Microb Cell Fact 14:182.
c. de la Torre A, et al. 2015. Genome-scale metabolic reconstructions and theoretical investigation of methane conversion in Methylomicrobium buryatense strain 5G(B1). Microb Cell Fact 14:188.
d. Lieven C, et al. 2018. A genome-scale metabolic model for Methylococcus capsulatus (Bath) suggests reduced efficiency electron transfer to the particulate methane monooxygenase.
e. Fu Y, et al. 2019. Core metabolism shifts during growth on methanol versus methane in the methanotroph Methylomicrobium buryatense 5GB1. mBio

2. For the type I methanotrophs, the ribulose monophosphate (RuMP) cycle should be the major route for formaldehyde assimilation. However, based on this study, this cycle remains inactive under all growth conditions for both strains: fluxes of H6P synthase are all 0.0 in figure 3-6, and the serine cycle is predicted to be the major carbon assimilation pathway in many cases. I don’t think this is correct. It has been reported that serine cycle carries a minimal flux during growth on methane for Methylomicrobium buryatense 5GB1C (Fu, et al. 2017). Although it can be more active in a different growth condition or in a different methanotroph, the serine cycle should not be the major pathway responsible for C1 assimilation in the type I methanotrophs.

In addition, the in vivo reaction rate of formaldehyde + THF->methylene-THF is reported to be low (Hai He, et al. 2020; Peyraud R, et al. 2011; Gregory J. Crowther, et al. 2007). But in your flux maps, this reaction is too active. Please note that this is not an enzymatic reaction but a spontaneous reaction. And evidence has shown that formate, instead of formaldehyde, is assimilated into the serine cycle (See ref c-d), although classic articles described formaldehyde as the branch point.

a. Fu Y, et al. 2017. The oxidative TCA cycle operates during methanotrophic growth of the type I methanotroph Methylomicrobium buryatense 5GB1. Metab Eng 42:43–51.
b. Hai He, et al. 2020. In Vivo Rate of Formaldehyde Condensation with Tetrahydrofolate. Metabolites, 10, 65.
c. Gregory J. Crowther, et al. 2007. Formate as the Main Branch Point for Methylotrophic Metabolism in Methylobacterium extorquens AM1. J. BACTERIOL.
d. Peyraud R, et al. 2011. Genome-scale reconstruction and system level investigation of the metabolic network of Methylobacterium extorquens AM1. BMC Syst Biol 5:189

3. I am not fully convinced by your prediction of the formaldehyde exchange between Methylobacter and Methylomonas. According to Ingeborg Bussmann, et al., “formaldehyde production (0.2–7 mM) by methanotrophs was reported only during growth at high methanol and oxygen concentrations”. However, you have reported production and exchange of formaldehyde under oxygen limitation (Figure 1B). Some experimental evidence has to be presented for further validation.
a. Ingeborg Bussmann, Monali Rahalkar & Bernhard Schink. Cultivation of methanotrophic bacteria in opposing gradients of methane and oxygen FEMS Microbiol Ecol 56 (2006) 331–344

Therefore, you have to further curate your models till the simulation makes physiological sense. Except the above papers, following papers may be useful to your study.

• Akberdin IR, et al. 2018. Methane utilization in Methylomicrobium alcaliphilum 20ZR: a systems approach. Sci Rep 8:2512.
• Gilman A, Fu Y, Hendershott M, Chu F, Puri AW, Smith AL, Pesesky M, Lieberman R, Beck DA, Lidstrom ME. 2017. Oxygen-limited metabolism in the methanotroph Methylomicrobium buryatense 5GB1C. PeerJ 5:e3945.

---

## Round 0.2 · Major Revisions

The two reviewers, who are experts in metabolic modeling, focused on this central part of the study.

However, from the biological point of view, there are many remaining weaknesses, even after revision.

Major concerns

I understand that you do not do wet lab experiments, so, by default, you cannot test whether your model predictions make any biological sense. You should probably state that upfront. However, you can also use the existing literature to at least make some correlations with the model predictions. In Fig. 2, under no limitations, Methylomonas constitutes about 30 % of the total community, while, in experimental synthetic communities, Methylomonas is completely dominant.

Your simulations definitely mimic synthetic and not natural communities, as factors other than input/output metabolites are missing, such as predators (they could be selectively affecting Methylomonas), other interacting species, etc.

I do not understand where the idea of formaldehyde comes from and why do you need it to be excreted? It seems like you have decided that formaldehyde should be a factor and randomly restrained your model in a predetermined way. Meantime the experimental evidence for excreted methanol (by both species) was ignored.

Other comments

L 35 Here and elsewhere, tundripaludum should not be capitalized

L 35 Multiple strains of Methylobacter are present in natural communities, not necessarily 21/22. It is better to use a general name without any strain name, in this context.

L 35 Methylomonas species are never major species in natural communities, only is synthetic communities. Strain LW13 is especially successful in outcompeting all others, a typical weed microbe. Its artificial fitness in the lab obviously does not translate into environmental fitness. Please present the correct information.

L 64 Actually, methanotrophs could be aerobic (Proteobacteria, Verrucomicrobia) or anaerobic (NC10 bacteria, ANME Archaea). Make sure you specify you are talking about aerobic bacteria, and perhaps mention existence of verrucomicrobial methantrophs briefly.

L 74 Same, specify that you are talking about aerobic proteobacterial methanotrophs.

LL 79-82. This classification is outdated. First, ALL Type I methanotrophs encode at least the majority of the serine cycle reactions. Second, there are no data on Methylococcus using the serine cycle any more than any other Type I methanotroph. Third, to me, type X is not a real type and this type belongs to outdated pre-genomics literature.

L 89 What do you mean by major? In natural communities, methanotrohs may be relatively abundant but they are definitely not major, relative to other functional groups.

L 99 Chloroplast means Cyanobacteria. Please say Cyanobacteria instead of ‘organisms containing chloroplast sequences’.

L 120 Better to specify ‘outcompeted by Methylomonas’. Many other members of the communities were there, some methanotrophs and some non-methanotrophs. Also consider additional data from a 2017 manuscript.
Natural Selection in Synthetic Communities Highlights the Roles of Methylococcaceae and Methylophilaceae and Suggests Differential Roles for Alternative Methanol Dehydrogenases in Methane Consumption.
Yu Z, Beck DAC, Chistoserdova L.
Front Microbiol. 2017 Dec 5;8:2392. doi: 10.3389/fmicb.2017.02392. eCollection 2017.


L 121 Isotope, not isotype.


L 122 Also consider additional data from an earlier manuscript

High-resolution metagenomics targets specific functional types in complex microbial communities.
Kalyuzhnaya MG, Lapidus A, Ivanova N, Copeland AC, McHardy AC, Szeto E, Salamov A, Grigoriev IV, Suciu D, Levine SR, Markowitz VM, Rigoutsos I, Tringe SG, Bruce DC, Richardson PM, Lidstrom ME, Chistoserdova L.
Nat Biotechnol. 2008 Sep;26(9):1029-34. doi: 10.1038/nbt.1488.

L 123 It was not Methylosinus, it was Methylosarcina. Please correct.

L 123 ‘Other species’, specify these were Methylomonas and Methylosarcina. If Methylosinus was included, perhaps the situation would be different.

L 159 Methylomonas is not naturally abundant based on sequencing communities from natural sediments.

L 165 Perhaps better to say coenzyme B12?

L 167, which known transported metabolites and which literature?

L 177 First, what data do you have on Methylomonas excreting and Methylobacter consuming formaldehyde? I am pretty sure they do not. Thus, how can you conclude on altruistic behavior? In natural communities, Methylobacter outcompetes Methylomonas, does this also make it altruistic? Then both are altruistic? On another hand, both are known to excrete methanol (Krause et al 2017 and reference below).

Physiological Effect of XoxG(4) on Lanthanide-Dependent Methanotrophy.
Zheng Y, Huang J, Zhao F, Chistoserdova L.
mBio. 2018 Mar 27;9(2):e02430-17. doi: 10.1128/mBio.02430-17.

L 181 I do not understand what you mean ‘natural’ versus ‘synthetic’ they are both synthetic. Natural communities consist of approximately 3000 species, and the interconnections are very complex beyond methane oxidation, carbon sharing, predation etc. Your simulations are absolutely artificial and thus cannot represent natural communities.

L 305 May or may not be. This random and old reference presents a very poor justification. Both organisms in that study may have been incorrectly identified anyway. Before methanotroph genomics, classification was very poor. Methylobacter and Methylomonas are famously intespersed in many phylogenetic trees. In addition, ALL methanotrophs, and life in general, are sensitive to formaldehyde. Formaldehyde is a potent toxin.

LL 328-330 These are very questionable choices of literature. Kalyuzhnaya et al 2015 is just a genome announcement, there is nothing about metabolic interactions. All others contain speculations or hypotheses, not data. What is known is that all methanotrophs appear to secrete methanol and other organics and share them with other community members. Of course, your system is a simplified superficial system, but, still, you seem to pick your answer from some outdated literature and then support it by modeling, instead of using your modeling effort to come to a certain answer.

L 365 Same question, what is your assumption based on? I do not believe formaldehyde is excreted or consumed in real life. Or at least you do not provide good support for such an assumption other than random references.

LL 392-393 How is it possible that only one species is affected by nutrient limitation? I have tons of experimental evidence that if you limit nutrients (methane, oxygen, nitrogen), ALL methanotrophs are affected, they simply stop growing and shut down gene transcription.

L 423 What is your evidence for ‘active’ serine cycle in Methylomonas but not Methylobacter?

L 425 How did you come to this conclusion?

L 430 Where does acetaldehyde come from? How is it converted into pyruvate? How is CO2 assimilated and why? Why would Methylobacter not do the same? None are known to be autotrophs.

LL 433, 547 How can you discuss enzyme activities when you have no experimental tests? To me this is all fantasy.

L 492 What does this mean, Methylobacter relies on formaldehyde. Why? I mean it does, but it produces tons itself, it does not need to rely on external?

L 575 I do not think Beck et al 2013 suggested anything about Methylomonas co-operativity.

L 586 Better say cycling, not re-cycling.

L 595 Betaproteobacteria ARE Proteobacteria.

LL 596-599 You are not trying to represent Lake Washington community it represents itself. Perhaps you are trying to say that a more complex model system will better represent metabolism of methane by a complex community?

Fig. 1. Note that both Methylobacter and Methylomonas are motile, by the means of a single flagellum. Methylobacter is more oblong, and Methylomonas is more round. Both are smooth, no spikes. People who know methanotrophs would assume that the curved forms are Methylosinus. Please use alternative graphics.

Fig. 7. I understand that, with adjusting various parameters, models can be adjusted to what you want them to be. However, the lack of CO2 under any condition makes no sense. Oxidation of methane will always produce CO2.

Reviewer 1 ·

Basic reporting

The manuscript is well written, clear and unambiguous. The introduction covered all the content needed to understand the results with the literature well-referenced and current. All figures are in good quality and well described.

Experimental design

This work is within the scope of the journal and shows a well-designed and rigorous investigation with coherent methods to achieve the objectives. Methods were described with sufficient detail and information to replicate.

Validity of the findings

The microbial community interactions of methanotrophs was appropriately analyzed using ­­­­­­­genome-scale metabolic modeling. The microbial community interactions of methanotrophs is of interest to the field, so this study is a welcome addition.

Additional comments

The authors have adequately addressed all concerns raised by reviewer. Thus, this revised manuscript can be accepted as is.

Reviewer 2 ·

Basic reporting

The modeling results make more sense after correction of several major issues. One minor issue of Fig 1.: the pictures of Methylomonas and Methylobacter could be misleading. They should not look like E. coli.

Experimental design

Materials and Methods part has provided detailed information of how the model was developed and manually curated. Most major issues, such as no flux through the RuMP cycle and a strong serine cycle, have been corrected. I have no other major concerns on the multi-objective optimization or how the simulation was carried out.

Validity of the findings

No comment.

Additional comments

One interesting finding from this study is that the ED pathway is the predominant glycolytic pathway. The EMP pathway shows little or no flux from G3P to PYR. However, many studies have shown the otherwise for the type I methanotrophs. Of course, no one has yet measured glycolytic fluxes experimentally in Methylobacter and Methylomonas species. The authors may want to mention this or give some explanation in the manuscript.

1) Kalyuzhnaya MG, et al. Highly efficient methane biocatalysis revealed in a methanotrophic bacterium. Nat Commun 4:1–7.
2) He L, et al. Quantifying methane and methanol metabolism of “Methylotuvimicrobium buryatense” 5GB1C under substrate limitation. mSystems 4:1–14
3) Torre A, et al. Genome-scale metabolic reconstructions and theoretical investigation of methane conversion in Methylomicrobium buryatense strain 5G(B1). Microb Cell Fact 14:1–15

---

## Round 0.3 · accepted · Accept

I think we have reached a consensus at this point. I think this manuscript presents a novel insight into metabolism of methane, especially as relevant to methanotroph communities, even if in purely in silico sense.